# Dark Current Modeling for a Polyimide—Amorphous Lead Oxide-Based Direct Conversion X-ray Detector

**DOI:** 10.3390/s22155829

**Published:** 2022-08-04

**Authors:** Tristen Thibault, Oleksandr Grynko, Emma Pineau, Alla Reznik

**Affiliations:** 1Department of Physics, Lakehead University, Thunder Bay, ON P7B 5E1, Canada; 2Thunder Bay Regional Health Research Institute, Thunder Bay, ON P7B 6V4, Canada

**Keywords:** amorphous lead oxide, blocking layer, mathematical model, dark current, direct conversion, kinetics, polyimide, X-ray detector

## Abstract

The reduction of the dark current (DC) to a tolerable level in amorphous selenium (a-Se) X-ray photoconductors was one of the key factors that led to the successful commercialization of a-Se-based direct conversion flat panel X-ray imagers (FPXIs) and their widespread clinical use. Here, we discuss the origin of DC in another X-ray photoconductive structure that utilizes amorphous lead oxide (a-PbO) as an X-ray-to-charge transducer and polyimide (PI) as a blocking layer. The transient DC in a PI/a-PbO detector is measured at different applied electric fields (5–20 V/μm). The experimental results are used to develop a theoretical model describing the electric field-dependent transient behavior of DC. The results of the DC kinetics modeling show that the DC, shortly after the bias application, is primarily controlled by the injection of holes from the positively biased electrode and gradually decays with time to a steady-state value. DC decays by the overarching mechanism of an electric field redistribution, caused by the accumulation of trapped holes in deep localized states within the bulk of PI. Thermal generation and subsequent multiple-trapping (MT) controlled transport of holes within the a-PbO layer governs the steady-state value at all the applied fields investigated here, except for the largest applied field of 20 V/μm. This suggests that a thicker layer of PI would be more optimal to suppress DC in the PI/a-PbO detector presented here. The model can be used to find an approximate optimal thickness of PI for future iterations of PI/a-PbO detectors without the need for time and labor-intensive experimental trial and error. In addition, we show that accounting for the field-induced charge carrier release from traps, enhanced by charge hopping transitions between the traps, yields an excellent fit between the experimental and simulated results, thus, clarifying the dynamic process of reaching a steady-state occupancy level of the deep localized states in the PI. Practically, the electric field redistribution causes the internal field to increase in magnitude in the a-PbO layer, thus improving charge collection efficiency and temporal performance over time, as confirmed by experimental results. The electric field redistribution can be implemented as a warm-up time for a-PbO-based detectors.

## 1. Introduction

Advanced direct conversion X-ray detectors utilize a layer of photoconductive material that acts as an X-ray-to-charge transducer, deposited over a large area imaging array [1,2]. Metal electrodes are used to establish an electric field within the photoconductor [3]. In the only commercially implemented direct conversion detectors, which are based on amorphous selenium (a-Se) photoconductors [1,4], a typical operating applied electric field is ≥10 V/μm. Such a strong electric field is needed to suppress mutual recombination of X-ray generated electron-hole pairs (ehp) and provide adequate charge collection efficiently. The dark current (DC) in such a metal/a-Se/metal photoconductive structure would be unacceptably large if preventative measures were not taken.

To suppress DC to acceptable levels (1–10 pA/mm^2^ depending on application [1,5]) in a-Se detectors, charge blocking layers are implemented [4,6,7]. Conventionally, a-Se photoconductive structures are multilayered, consisting of an intrinsic photoconductive layer of a-Se (*i*-layer) sandwiched between two charge blocking layers of doped or alloyed a-Se [6]: a thin (a few µm) alkali metal-doped a-Se layer blocks the injection of holes and allows the passage of photogenerated electrons, denoted as the *n*-like layer [8]; and a few µm thick As_2_Se_3_ *p*-like blocking layer blocks the injection of electrons and allows the passage of photogenerated holes [4].

Although practically used in a-Se detectors, *p*-like and *n*-like layers are technically complex to produce, requiring a co-thermal vacuum deposition process [9]. In addition, *p*-like and *n*-like blocking layers are less effective in suppressing DC when detectors are biased at fields >10 V/µm, which is needed to improve charge collection efficiency [10]. Recent research has shown that certain polymers (polyimide (PI) [11], cellulose acetate (CA) [10], and perylene tetracarboxylic bisbenzimidazole (PTCBI) [12]) are promising candidates as blocking layers. These materials maintain effectiveness when detectors are biased at high fields, are technologically less complex to produce, and are compatible with large area flat panel detector technologies. That is, they can be coated uniformly on a large area imaging array so that the photoconductive layer is subsequently deposited directly onto blocking layers [10,12,13]. Among these potential candidates, PI stands out as it demonstrated its effectiveness when incorporated into practical a-Se detector structures: PI can maintain a sufficiently low DC (less than 10 pA/mm^2^) even under a high electric field where impact ionization occurs in a-Se [14]. It was also shown that a PI blocking layer aids in the suppression of signal lag in a-Se detectors [10,12]. Despite these encouraging results, the exact transport properties of PI interfaced with photoconductors in a detector structure are not well understood [15], and concerns remain that interfacing a foreign material such as PI with a photoconductor other than a-Se, could affect detector performance. Although it is very tempting to use the PI blocking layer in conjunction with an amorphous lead oxide (a-PbO) photoconductor [5,16,17,18], it is challenging to predict the level of DC that will flow through a PI/a-PbO detector at a given time and electric field. Moreover, other factors affecting DC, such as possible charge accumulation at the PI/a-PbO interface, are needed to be understood. Furthermore, the PI/a-PbO detector must maintain the fast X-ray response inherent to a-PbO detectors [19] to ensure its feasibility for real-time imaging applications. Therefore, it is imperative to verify the low-lag (low residual signal after the termination of X-ray exposure) operation of the multilayered PI/a-PbO detector.

In this work, an a-PbO-based single-pixel detector prototype with a thin layer of PI (a “thin” layer refers to the fact that the blocking layer thickness is much smaller than the photoconductor thickness) positioned between the bottom electrode and the a-PbO layer is investigated in terms of the temporal performance, sensitivity, and bias-dependent transient behavior of DC. DC kinetics in a PI/a-PbO detector is simulated via a mathematical model and fitted to experimental data to understand the kinetics and processes that govern the suppression of DC in PI/a-PbO photoconductive structures.

## 2. Materials and Methods

### 2.1. Dector Fabrication

A single-pixel PI/a-PbO X-ray detector [5,16] has been fabricated as the prototype for an a-PbO-based direct conversion flat panel imager. In Figure 1a, a schematic diagram of the detector prototype structure is illustrated, and in Figure 1b, an SEM cross-sectional image is shown. It was made in very few numbers of processing steps that included the use of a PI blocking layer. An indium-tin-oxide (ITO) coated glass slide was used as a substrate and a bottom electrode; it was cleaned thoroughly with acetone, methanol, and isopropanol and dried under N_2_. A 1.1 µm thick PI layer was deposited on it by spin coating. A mask was utilized during the spin coating of PI to ensure a small area of ITO remains uncoated for the later purpose of electrical connection, essential for DC kinetics experiments. An 18.5 μm layer of a-PbO was then deposited using ion-assisted thermal deposition [5,20] onto the PI-coated substrate. A gold contact of area ≈ 1 mm^2^ and thickness of 20 nm was sputtered on top of the a-PbO, forming the top electrode. The smallest electrode determines the effective area of the detector, i.e., the Au contact; thus, the detector’s pixel size is ≈ 1 mm^2^. Previous publications [5,20] contain a detailed description of the ITO substrate preparation, PI layer spin-coating, and a-PbO layer ion-assisted thermal deposition processes.

### 2.2. Experimental Setup—Temporal Performance and Sensitivity Characterization

Evaluation of temporal performance was conducted using an X-ray-induced photocurrent method (XPM) in continuous and modulated modes. In both modes, a biased single-pixel PI/a-PbO X-ray detector was exposed to either continuous or modulated X-ray pulses, and an X-ray-induced photocurrent was evaluated. The modulated mode of XPM was used to calculate signal lag [5]. Figure 2a shows a schematic of an experimental setup for a modulated XPM, where a PI/a-PbO detector is exposed to a sequence of short ≈ 16.67 ms X-ray pulses with a ≈ 16.67 ms interval between them. The pulsed irradiation was achieved by modulating a continuous X-ray pulse generated with a radiographic X-ray unit with a 0.3 mm thick copper chopper. The frequency of the chopper’s modulation was 30 Hz, corresponding to the frame rate of 30 frames per second (fps) commonly used in fluoroscopy [21].

The continuous mode of XPM was used to evaluate detector sensitivity. Sensitivity to X-rays is characterized in terms of energy required to create a single detectable ehp (W_±_). W_±_ is derived from the total charge collected upon continuous X-ray exposure: the lower W_±_ is, the higher is sensitivity. More details on this analysis can be found elsewhere [16,22,23]. In a-PbO, W_±_ was found to depend on the applied electric field. As discussed later, it is believed that the electric field undergoes a redistribution within the PI/a-PbO detector after the application of a bias. Therefore, W_±_ was measured by a series of successive pulses immediately after the application of a bias and compared to W_±_ measured 10 min after bias application. In both cases (immediate irradiation and irradiation after 10 min), the sample was allowed to rest in a short-circuited fashion, releasing any previously trapped charge. The experimental configuration, seen in Figure 2a, was used for W_±_ measurements with the condition that the chopper was fixed in the open state, allowing a continuous beam of irradiation to pass.

The PI/a-PbO detector was housed within an Al box and biased in positive polarity (positive voltage applied to the ITO contact) to a field of 20 V/μm with a Stanford Research Systems PS350 power supply. To achieve the desired field (*F*_0_), a bias of (*V*_0_ = *F*_0_ × (*L_PbO_ + L_PI_*)) was applied to the sample, where *L_PbO_* is the thickness of the a-PbO layer and *L_PI_* is the thickness of the PI layer. The photocurrent was read out from the top Au contact with a Tektronix TDS 2024C oscilloscope with a 1 MΩ native input resistance. The sample was biased for 10 min before X-ray exposure, apart from W_±_ measurements immediately after bias application. The beam of X-rays was produced by a Dunlee PX1412CS tube with a DU-304 insert and a tungsten target. A 200 ms long, 60 kVp pulse of X-rays passed through 1.3 mm of Al filtration, which hardened the beam, and then was collimated by a 2 mm thick lead plate. The resulting poly-energetic X-ray beam incident on the detector has an average energy of ≈ 36 keV with energies ranging from 11 keV to 60 keV.

### 2.3. Experimental Setup—DC Kinetics

DC kinetics was measured for various applied fields (*F*_0_ = 5, 10, 15, and 20 V/μm), conventional for efficient detector operation. The experimental setup can be seen in Figure 2b, where a Stanford Research Systems PS350 power supply applies a positive bias to the ITO contact. A Keithley 35617EBS electrometer connected to the top Au contact measured the resultant DC. The electrometer and the power supply were controlled via a script executed on a control computer, connected by a GPIB interface (Tektronix AD007). The power supply was ramped to *V*_0_ at 5 V/s to avoid large magnitude spikes in DC that could damage the detector. Once *F*_0_ was achieved, the electrometer recorded the DC at a rate of 1 s^−1^ for two hours. After the DC recording period passed, the bias voltage ramped to 0 V at the same rate, and the detector was held in a resting configuration for 4 h. A 100 MΩ resistor in parallel with the detector ensured proper grounding during the resting period. The detector was installed within a light-tight box, which prevents any photogenerated current, and allowed to rest in a short-circuit fashion for several hours before the experiment began to drain any previously trapped charge.

## 3. Results

### 3.1. Temporal Performance and Sensitivity Characterization

Figure 3 compares the photocurrents induced by continuous irradiation for 0.2 s and by modulated irradiation at 30 frames per second of the PI/a-PbO detector. The photocurrents are normalized to the steady-state magnitude of the continuous response. When the PI/a-PbO detector was exposed to continuous irradiation, it demonstrated a quasi-rectangular X-ray response. At the beginning of irradiation, the photocurrent increased almost instantly and remained at a nearly constant amplitude throughout the X-ray pulse duration. After the termination of the X-ray pulses, the amplitude rapidly fell back to the DC level. A slight increase in photocurrent at the start of the X-ray response is due to the characteristic overshoot of the X-ray flux rather than the detector’s behavior, as verified by an identical response from a silicon photodiode.

The response of the PI/a-PbO detector to a series of short X-ray pulses with a rate of 30 frames per second indicates excellent temporal performance. During each frame, the photocurrent rises to the level of the continuous pulse response and remains there for the frame’s duration. The response drops almost to the DC level when the frame ends. Lag values were calculated using the concept described in [5,19,22] and were found to be ~1%.

Figure 4 shows how ehp creation energy decreases with time after bias is applied. It was measured sequentially immediately after applying the bias (denoted in Figure 4 as $W_{+}^{\mathrm{inst}.}$), with a short interval of 10 s between adjacent measurements. The obtained values were normalized to the reference point, which is W± measured 10 min after applying the bias. Figure 4 presents only the measurements acquired during the first ~40 s after applying the bias and shows that initially, $W_{+}^{\mathrm{inst}.}$ is ~30% greater than the reference point and approaches it over time.

### 3.2. DC Kinetics

Figure 5 shows experimental DC kinetics data plotted in a semi-log scale at different fields. DC decays by almost two orders of magnitude post-bias application. At 10 V/μm, the most relevant field for direct conversion detector operation, DC magnitude is initially ≈26 pA/mm^2^ and decays to ≈0.3 pA/mm^2^ two hours after bias application. With an increased applied field, the magnitude of DC increases. However, at all fields, DC decays below the operational threshold of 1 pA/mm^2^ after two hours, as seen in Figure 5.

### 3.3. Mathematical Model

A mathematical model was derived to probe the DC decay mechanisms present in the PI/a-PbO detector by simulating experimental kinetics data. Our model uses an approach developed in [6,15,24,25] to simulate DC kinetics in a-Se blocking structures and extend it to account for the peculiarities of PI/a-PbO detectors. The model is based upon the following assumptions: (1) The primary source of DC in the PI/a-PbO detector is the injection of holes from the positively biased electrode proportional to the electric field at the electrode/PI interface. This is the case for a-Se detectors [15,24,25], which exhibit similar DC behavior compared to a-PbO-based detectors. The assumption that injection is the primary source of DC is further fortified by the fact that the inclusion of a blocking layer, engineered to suppress injection, reduces DC significantly in a-PbO-based detectors. A secondary source of DC is the thermal generation and subsequent multiple-trapping (MT) controlled transport in the bulk of a-PbO. The injection and thermal generation of holes are exclusively considered as they are major carriers in PbO [26,27]. (2) Injected holes are deeply trapped within the PI layer, which screens the applied electric field, a mechanism established for PI blocking layers in [15]. The resulting field redistribution within the detector structure occurs, causing the magnitude of the field at the ITO/PI interface to decrease, reducing injection and DC. (3) The concentration of deep trapping states within the a-PbO layer is negligible compared to that in PI [15,28], and thus trapping in the bulk of a-PbO can be negated in terms of space charge accumulation that would contribute to the field redistribution.

The time-dependent electric displacement field redistribution and MT transport are illustrated in Figure 6, where *F_PI_*(*x*,*t*) and *F_PbO_*(*t*) are the electric fields within the PI and a-PbO layers, respectively; *ε*_0_ is the vacuum permittivity, *ε_r,PI_* is the relative permittivity of PI, *ε_r,PbO_* is the relative permittivity of a-PbO, and *E_V_* is the valence mobility edge of a-PbO. The exact expressions for *F_PI_*(*x*,*t*) and *F_PbO_*(*t*) will be discussed and derived later in the text (Equations (5) and (8)).

In the model, holes undergo thermionic emission over a Schottky potential barrier, from the positively biased electrode (ITO) into the PI layer. The resulting current density of holes is described by:(1)$$J_{h}\left(t\right)=eN_{v}\mu_{h}F_{PI}\left(0,t\right)exp\left[\frac{-\left(\varphi_{h}-\beta_{S}\sqrt{F_{PI}\left(0,t\right)}\right)}{kT}\right],$$
where *e* is the elementary charge, $N_{v}$ is the effective concentration of states in the valence band, $\mu_{h}$ is the effective hole mobility, $\varphi_{h}$ is the energy barrier height experienced by holes in the absence of an applied field, and *kT* is thermal energy, all pertaining to PI. In addition, $\beta_{S}$ is the Schottky coefficient that is equal to $\sqrt{e^{3}/\left(4\pi \varepsilon_{r,PI}\varepsilon_{0}\right)}.$

The drift of the injected holes induces a current density of $J_{h}\left(t\right)=e\mu_{h}p\left(t\right)F_{PI}\left(0,t\right)$ where *p*(*t*) is the concentration of drifting holes. Therefore, the concentration of drifting holes near the ITO/PI interface is described by:(2)$$p\left(t\right)=N_{v}exp\left[\frac{-\left(\varphi_{h}-\beta_{S}\sqrt{F_{PI}\left(0,t\right)}\right)}{kT}\right].$$

Drifting holes are then captured by energetically distributed localized states in PI. Trapped holes can be released after some time, proportional to an activation energy needed to escape, and be re-trapped into other unoccupied states. This model considers a uniform volume concentration of trapping states within the bulk of PI ($N_{\rho ,m}$). The trapping levels have been segmented in *m* discrete energy depths within the bandgap of PI. The differential equations to describe the trapping rate of holes within the bulk of PI are:(3)$$\frac{d\rho_{m}\left(t\right)}{dt}=p\left(t\right)C_{t,m}\left[N_{\rho ,m}-\rho_{m}\left(t\right)\right]-\frac{\rho_{m}\left(t\right)}{\tau_{r,m}},$$

Here, $C_{t,m}$ is the deep trapping coefficient related to the trapping time constant (*τ_c_*) by $C_{t,m}=1/(N_{\rho ,m}\tau_{c})$ [29]. Additionally, $\tau_{r,m}$ is the release time constant, which is exponentially dependent on the activation energy $\left(E_{a,PI}\right)$ needed for a hole to escape from a trap by:(4)$$\tau_{r,m}=\frac{1}{\omega_{0}}\exp\left[\frac{E_{a,PI}}{kT}\right],$$
where $1/\omega_{0}$ is the pre-exponential factor and *ω*_0_ is usually assumed to be on the order of the phonon frequency. Without an electric field, the activation energy needed for a hole to escape from an individual trap in Equation (4) is equal to the energy or depth ($E_{\rho ,m})$ of a trap. However, when an electric field is applied, thermally assisted tunneling lowers the activation barrier for a hole to be released from the trap. As a result, $E_{a,PI}$ in Equation (4) becomes smaller than the trap depth by $E_{a,PI}=E_{\rho ,m}-\Delta E=E_{\rho ,m}-e\beta F_{PI}\left(0,t\right)$, where β is proportional to the tunneling distance under the energy barrier [30]. As for the *τ_c_*, it is treated as an electric field independent constant as a first-level approximation.

The instantaneous electric field at the ITO/PI interface is found by solving Poisson’s equation in 1D cartesian coordinates, with the following boundary conditions: The integral of the electric field distribution must be equal to *V*_0_, the potential is continuous at the PI/a-PbO interface, and the displacement electric field at the PI/a-PbO interface is continuous. Solving for the field at the ITO/PI interface, the following expression was obtained:(5)$$F_{PI}\left(0,t\right)=\frac{V_{0}}{\left(\frac{\varepsilon_{r,PI}}{\varepsilon_{r,PbO}}L_{PbO}+L_{PI}\right)}-\frac{e\rho \left(t\right)L_{PI}}{2\varepsilon_{r,PI}\varepsilon_{0}}\left(\frac{\varepsilon_{r,PbO}L_{PI}+2\varepsilon_{r,PI}L_{PbO}}{\varepsilon_{r,PbO}L_{PI}+\varepsilon_{r,PI}L_{PbO}}\right)$$
(6)$$\rho \left(t\right)=\underset{m}{\sum }\rho_{m}\left(t\right),$$
where ρ(t) is the volume density of trapped holes within the bulk of PI. Here, as the density of trapped charge increases, the applied electric field is screened, and the field at the ITO/PI interface is lowered. As the field decreases at the interface, so does injection.

Another component of DC is the generation and transport of equilibrium holes in the bulk of a-PbO. Here we assume that holes drift through the bulk of a-PbO by MT mechanisms which is commonly considered in inorganic disordered semiconductors [30]. In the MT process, holes move only via extended states below the valence mobility edge. This motion is interrupted by the trapping of carriers into shallow localized states within the band tails and subsequently undergoes field-assisted thermal release back into extended states [31]. To account for the transport mechanisms present in the MT regime, an effective temperature (*T_eff_*) is introduced [31], given by:(7)$$T_{eff}={\left[T^{2}+{\left(\gamma \frac{eF_{PbO}\left(t\right)a}{k}\right)}^{2}\right]}^{1/2}.$$

In Equation (7), γ is a dimensionless coefficient and *a* is the localization length of trapping states in the band tail. The field within the bulk of a-PbO (*F_PbO_*(*t*)) is similarly derived as *F_PI_*(*x*,*t*) was and is given by:(8)$$F_{PbO}\left(t\right)=\frac{V_{0}}{\left(\frac{\varepsilon_{r,PbO}}{\varepsilon_{r,PI}}L_{PI}+L_{PbO}\right)}+\frac{e\rho \left(t\right)L_{PI}{}^{2}}{2\varepsilon_{0}\left(\varepsilon_{r,PbO}L_{PI}+\varepsilon_{r,PI}L_{PbO}\right)}.$$

In the MT transport regime, mobility is dependent on *T_eff_.* The effective temperature-dependent hole mobility in a-PbO is described as:(9)$$\mu_{h,PbO}\left(T_{eff}\right)=\mu_{0}exp\left[\frac{-E_{a,PbO}}{kT_{eff}}\right],$$
where *E_a,PbO_* is the average activation energy needed for a charge to escape from shallow localized states within the band tail.

To derive the current density induced by thermally generated holes, the concentration of drifting holes is found by solving the continuity equation for a uniform generation of holes throughout the bulk of a-PbO under steady-state conditions. Utilizing the solution to the continuity equation and $\mu_{h,PbO}\left(T_{eff}\right)$, the current density is:(10)$$J_{th}\left(t\right)=e\mu_{h,PbO}\left(T_{eff}\right)F_{PbO}\left(t\right)\tau_{h}g_{h}*\;\left[1-\frac{\mu_{h,PbO}\left(T_{eff}\right)F_{PbO}\left(t\right)\tau_{h}}{L_{PbO}}\left(1-exp\left[\frac{-L_{PbO}}{\mu_{h,PbO}\left(T_{eff}\right)F_{PbO}\left(t\right)\tau_{h}}\right]\right)\right].$$
where $\tau_{h}$ is the hole lifetime and $g_{h}$ is the thermal generation rate of holes in the bulk of a-PbO.

Thus, the total current density is:(11)$$J_{total}\;\left(t\right)=J_{h}\left(t\right)+J_{th}\left(t\right).$$

Field-dependent decay of DC was simulated by simultaneously solving the coupled first-order differential equations 3 numerically with Python. The model simplifies energetic disorder in PI by assuming three discrete trapping levels in the bulk of PI (see Table 1). Three levels are chosen as a first-level approximation, using the lowest number of levels that still accurately simulates the data. The range of trapping depths corresponds to previously reported results for PI [28]. The effective mobility of holes in PI is $\mu_{h}=$ 1 × 10^−6^ ${\mathrm{cm}}^{2}⁄\left(\mathrm{Vs}\right)$ [15], and the density of states within the valence band of PI is $N_{v}=6\times 10^{21}\;{\mathrm{cm}}^{-3}$ [15]. A typical phonon frequency was used equal to $\omega_{0}=1\times 10^{12}\;s^{-1}$ [6,9]. Other parameters such as the bulk concentrations of trapping states ($N_{\rho ,m}$) and intrinsic barrier height (*φ_h_*) are deposition dependent and therefore are treated as fitting parameters. $g_{h}$, *τ_c_*, and *β* are additionally treated as fitting parameters.

The relative dielectric permittivity of a-PbO and PI were determined by capacitance measurements through charge extraction by linearly increasing voltage (CELIV) without photoexcitation, i.e., dark-CELIV. Details pertaining to the technique of CELIV can be found in [18,26,27]. The dielectric permittivity of PI ($\varepsilon_{r,PI}=3.3$) and a-PbO ($\varepsilon_{r,PbO}=26$) was found. Measurements corresponded perfectly with manufacturer-specified dielectric permittivity of PI [32].

All parameters used in the model are shown in Table 1, and the simulation results can be seen in Figure 7 as dashed red lines. The simulated DC is broken into injected (dashed-dotted green lines) and thermally generated (blue lines) components. The model simultaneously simulated four different kinetics data corresponding to different applied fields. Meaning that all the fitting parameters are held constant throughout different applied fields, leaving only the parameter *V*_0_ to vary between them. Simultaneously fitting all four experimental data was deliberately chosen to adhere to physical accuracy since the parameters $N_{\rho ,m}$, *Ф_h_*, *τ_c_*, *β*, and $g_{h}$ in the PI/a-PbO detector have no dependence on the applied bias. As a result of fitting all four applied fields simultaneously, the fitting was guided by the overall average quality of fitting of all four datasets rather than the quality of fitting for each applied field individually. Overall, the fitting most accurately represents the experimental data (black lines) at higher applied fields of 15 and 20 V/µm, while accuracy decreased with decreasing applied fields. Generally, fitting is least accurate at short times, and as the simulation evolves, the fitting becomes more accurate. For each applied field, the initial DC due to hole injection is much larger than that due to thermally generated holes. As the field undergoes a redistribution, injection decreases until thermal generation becomes dominant, except for 20 V/µm.

## 4. Discussion

The experimental results demonstrate that the DC in the evaluated single-pixel PI/a-PbO prototype detector is a function of the applied bias and time: it decays with time from the instant of application of the nominal voltage and settles at low steady-state values. The steady-state value of DC depends on the applied voltage; however, for voltages relevant to the operation of the direct conversion flat-panel X-ray imagers (10–20 V/µm), it does not exceed 1 pA/mm^2^. This value lies in the low end of acceptable DC levels that is often quoted to be between 1 to 10 pA/mm^2^, depending on the exact application [1]. It is important to note that even at the highest applied field, tested here (20 V/μm), the DC in the presented PI/a-PbO detector prototype is on the same order of magnitude as in multilayer a-Se-based detectors at the same nominal applied field [14,15,35,36]. This suggests the possibility of using a higher field (i.e., 10 < *F*_0_ ≤ 20 V/µm rather than the 10 V/µm used in a-Se detectors) to improve charge collection and W_±_ while keeping DC at tolerable levels. In turn, the thickness of the PI layer should be optimized for operation at 20 V/µm. “Optimizing” means that the blocking layer should have adequate thickness to reduce the DC but not too thick so that a substantial fraction of the applied voltage drops over it rather than the photoconductor, as previously shown in [15]. This reduces the field inside the photoconductor, negatively affecting charge collection efficiency and the temporal performance of the detector.

Qualitatively, the mechanism of DC decay can be explained by deep charge carrier trapping and polarization effects that cause the instantaneous electric field at the ITO/PI interface to decrease and increase within the bulk of a-PbO. As a result, we achieve two very useful effects: on the one hand, the DC decreases, and on the other hand, the collection efficiency of X-ray generated charge is improved, and W_±_ decreases (since it is field-dependent [16,19]). This is shown in Figure 4, where W_±_ decreases after the instant of bias application. We visualize the field redistribution within the PI/a-PbO detector in Figure 8 where the electric displacement field ($D\left(t\right)=\varepsilon_{0}\varepsilon_{r}F\left(t\right)$) profile throughout the PI/a-PbO detector structure is plotted as a function of time for an applied electric field of 20 V/μm. The *D* profile is displayed in Figure 8 as the electric field profile is discontinuous at the PI/a-PbO interface due to the difference in the dielectric permittivity of PI and a-PbO. To avoid this discontinuity that would complicate visual interpretation, *D*, which is continuous, is displayed.

For a quantitative analysis of the bias-dependent transient DC, we assume that trapping occurs within the PI blocking layer itself. Of course, localized states have continuous distribution (most probably, an exponential density of states (DOS) typical for organic disordered materials). However, since the DOS is unknown, we replace it with a set of three discrete levels in the bulk (where the lower concentration of states corresponds to deeper centers). Despite the simplification, our model allows us to understand how much net space charge due to trapped carriers (e.g., holes) is needed to modify the internal field and to limit the injection of holes from the ITO electrode.

Numerical calculations show that within the simplifications of the presented model, it is impossible to find such a set of fitting parameters that provides a precise description of the experimental data in the entire time range tested here for all applied fields simultaneously. One can see from Figure 7 that the dashed red line perfectly agrees with experimental data at times near the end of the testing period but deviates from it at times shortly after bias application. Interestingly, the degrading quality of fitting in the initial time interval was also observed in [6,25], where simulated DC kinetics have been fitted to experimental kinetics at varying applied fields for an *n-i-p* a-Se detector. The good agreement between experimentally measured and simulated results at times when the DC decay begins to saturate is not surprising since it is ultimately the total amount of trapped charge, regardless of the mechanism of its trapping and distribution over localized states, that determines the steady-state DC. The situation is directly opposite at the initial stage of the field application when it is the probability and capture rate, the possible subsequent release of the trapped charge, and its redistribution over various energetically distributed localized states that determine the kinetics of DC. Although in our modeling, we made a step forward (in comparison to previous works on a-Se) to account for a field-assisted release from deep traps in the PI layer, the observed discrepancy between simulated and experimental results where experimental DC decays at a different rate than the simulated one suggests that there is an additional mechanism that affects the kinetics of the electric field redistribution, and it escapes our attention. Below we consider the hopping exchange of carriers between the traps and show how taking this effect into account improves the quality of the simulation even with a simplified energetic disorder in the bulk of PI, which assumes only three discrete deep trapping levels.

As was shown in [30], the release of charge from traps by thermally assisted tunneling can be further enhanced by field-assisted hopping transition from a given trap to a shallower surrounding trap. Since this process makes it easier for a hole to be activated to the valence band, the release times $\tau_{r,m}$ become shorter. In addition to the strength of the electric field, two factors influence the impact of this effect: the depth of a trap and the parameter $N_{\rho ,m}a^{3}$ [30]. For first-order approximations for these numbers, we used data from Table 1 and estimated $N_{\rho ,m}a^{3}$ to vary from 0.005 for the deepest trap of 1.0 eV (*m* = 3) to 0.18 for the shallowest trap of 0.82 eV (*m* = 1). The enhancement factor for the release of carriers from traps is defined as the ratio between the mean release time from a single trap and the release time from the same trap in the presence of a nearby trap at the optimal position [30]. This factor varies from ~2 for the shallowest trap at the weakest nominal field of 5 V/μm to ~9 for the deepest trap of 1.0 eV at the strongest nominal field of 20 V/μm. These estimations suggest that we should not neglect that a hole’s release from a trap to the valence band can be substantially enhanced by the presence of an additional, shallower trap. The hopping exchange of carriers between traps also provides additional channels for the capture of a hole from the valence band to a particular trap that leads to enhancement of the capture rate for a given trap [30]. Although the above considerations suggest that both $\tau_{c}$ and $\tau_{r,m}$ should be electric field-dependent, without knowing the exact concentrations and energy distribution of the localized states as well as the localization length and effective mass of carriers within these localized states, it is difficult for us to derive analytic equations describing the field-enhancement on the de-trapping process and the influence of the field on the capture probability. Instead, Equations (1)–(11) are left unmodified, and we let $\tau_{c}$ vary with the applied electric field and *β*, and therefore $\tau_{r,m}$, vary with trapping level depth within the band tail of PI. The field dependencies of $\tau_{c}$ and $\tau_{r,m}$ are derived from the best fit between the calculated and experimentally measured DC kinetics. The results are shown in Figure 9, Figure 10 and Figure 11.

Figure 9 shows the release times for the three discrete levels of traps with the energy depths of 0.82 eV, 0.86 eV, and 1 eV considered in our model. Much steeper field dependence for the deepest trap suggests that the release rate from this trap is enhanced by the presence of other (shallower) traps in full agreement with a model presented in [30]. This effect is accompanied by the equal enhancement of the capture time for the deepest trap due to the assistance of the surrounding traps. The dependence of such an enhancement on capture time is shown in Figure 10.

Figure 11 presents a comparison between experimental (solid lines) and simulated (dashed lines) DC kinetics data, modified to account for the hopping-assisted release of charge carriers from trapping sites. Much better agreement between the experimental and simulated results is evident when compared to Figure 7 (a 33.6% reduction in the total cumulative residual sum of squares (RSS) of all four applied fields).

It is also useful to examine the simulated kinetics of trap occupancy (Figure 12). Holes trapped within deep sites are effectively lost to conduction as the release time for these traps is longer than the duration of DC decay measurements. The other shallower levels (0.82 and 0.86 eV) play a role in decay, but holes trapped in these levels are released during the time frame of DC decay and then are lost to the deepest traps. Therefore, the major contribution to the creation of this positive space-charge barrier is made by charge trapped in the deepest traps (1 eV).

The total concentration of the deepest traps within a thickness of 1.1 μm of PI used here seems optimal when 10 V/μm is utilized as a nominal operating electric field. Indeed, this is seen in Figure 7; at 10 V/μm, the steady-state DC is dominated by thermal generation, not injection. However, a thicker layer of PI may be desired for larger electric fields to reduce the DC to a limit determined by thermal generation. Based on our simulated results, at 20 V/μm, this can be achieved with a PI layer thickness of 1.3 μm.

As previously mentioned, when introducing a foreign material (PI) into the structure of a direct conversion detector, the fear that states are created at the interface of the photoconductor and the foreign material is always present. During the model’s derivation, trapping states at the interface of PI and a-PbO have been included in previous iterations. However, the delineation between surface trapping and bulk trapping in PI does not change the fitting quality. The experimentally measured decay can be explained by the trapping in the bulk of PI alone. This indicates that the concentration of states at the interface is negligible compared to that in the bulk of PI.

Evaluation of temporal response to X-ray irradiation in the PI/a-PbO detector was very important to demonstrate that while suppressing DC to acceptable levels, the presence of the PI blocking layer does not degrade the performance of the PI/a-PbO detector. As can be seen in Figure 3, the amplitude of the response to a continuous beam of X-rays stays constant during irradiation, confirming three findings: (1) The steady-state redistribution of the electric field that is responsible for bias-dependent DC behavior is unaffected by the presence of the photogenerated charge. (2) Photogenerated charge does not experience deep trapping at the PI/a-PbO interface during the drift through the PI layer. (3) There is no X-ray-triggered injection occurring in the detector.

During irradiation of the PI/a-PbO detector to a modulated beam of X-rays, the photocurrent within each frame and the amplitude of successive frames remain constant. This contrasts with similar PI/a-Se detectors, whose photo-response to a beam of modulated X-rays exhibits unstable temporal response. The photo-response shown in [15] demonstrates that the photocurrent during each frame decreases over time. This behavior is attributed to the accumulation of photogenerated electrons at the PI/a-Se interface that temporarily degrades the internal electric field, restored via injection before the next frame begins [15]. In another publication, [14], pertaining to a PI/a-Se detector, the amplitude in each successive frame is shifted up. This behavior is caused by a rise in the DC level, speculated to result from increased injection triggered by the trapping of X-ray-generated electrons at the interface [14]. The response seen in Figure 3 indicates that the PI/a-PbO detector lacks these imperfections (i.e., interface states that trap photogenerated electrons), causing unstable temporal X-ray response in PI/a-Se detectors. Overall, the PI/a-PbO detector’s response to continuous and modulated X-ray pulses confirms that PI does not degrade temporal performance or charge collection efficiency.

## 5. Conclusions

This investigation reports on the temporal performance, sensitivity, and DC behavior of a PI/a-PbO detector. Characterization of temporal performance shows that it is unhindered by the inclusion of a PI layer, and the detector exhibits a lag of ~1% at 30 Hz and 20 V/μm. DC kinetics shows that at every field reported here (5–20 V/µm), a PI layer thickness of 1.1 µm was sufficient to cause DC to decay below the 1 pA/mm^2^ operational threshold. This reveals that PI is a practical approach to suppress DC in a-PbO-based direct conversion detectors. If necessary, the DC can be further reduced by simply increasing the thickness of the PI layer. Increasing the thickness of the PI layer will increase the total number of deep trapping centers that can accumulate space charge to suppress carrier injection further and stabilize the DC at the fundamental limit determined by thermal generation, even at large applied fields.

In order to explain the experimentally measured DC decay in the PI/a-PbO detector, in the entire range of electric fields accessed in this work, we modified a ‘standard’ mathematical model used to simulate DC kinetics derived by others [6,15,24,25] to account for hopping-assisted release and capture of holes with filed-dependent trapping and release time constants. A noticeable improvement in the quality of fitting with the modified model, in comparison with a standard model, indicates that hopping transitions of trapped holes between localized states in the band tail substantially influences both the field-induced hole release and trapping. Both mechanisms play essential roles in reaching a steady-state occupancy level of different traps in the bulk of PI, resulting in a particular profile of the internal electric field.

In addition to the above theoretical considerations, our results suggest three practical results. First, a basic set of algebraic equations can be applied to find the numerical solution to the kinetics of electric field evolution that includes a realistic process of charge trapping. Secondly, the model can be used to approximate the optimal PI thickness for future iterations of a-PbO-based detectors without the need for time- and labor-intensive experimental trial and error. This can be applied to direct conversion detectors based on other photoconductors interfaced with PI by modifying the presented model to account for the particularities of the photoconductor. Finally, sensitivity measurements presented here revealed that a field redistribution and subsequent field increase within the a-PbO layer improves the detector’s photogenerated charge collection efficiency and temporal performance. This can be implemented as a warm-up time, a common requirement of sophisticated electronics, for a-PbO-based detectors to improve SNR. Overall, comprehending DC mechanisms of direct conversion detectors utilizing foreign material blocking layers, such as PI, gives essential insight into improving and optimizing their development.

## Figures and Tables

**Figure 1 sensors-22-05829-f001:**
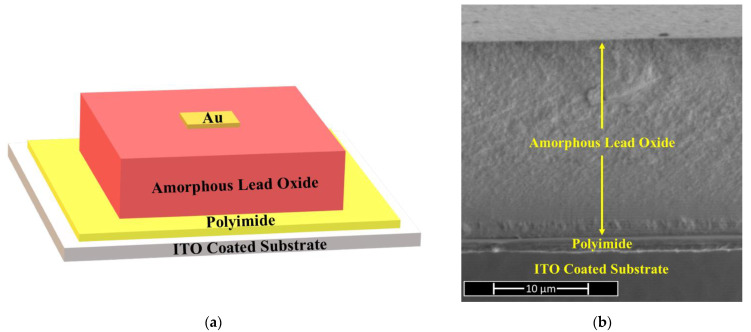
(**a**) Schematic diagram (not to scale) and (**b**) a cross-sectional scanning electron microscopy (SEM) image of a single-pixel PI/a-PbO direct conversion X-ray detector.

**Figure 2 sensors-22-05829-f002:**
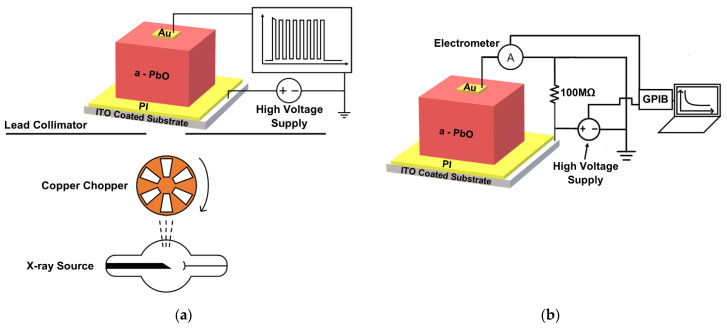
Experimental setups for (**a**) modulated and continuous XPM and (**b**) DC kinetics.

**Figure 3 sensors-22-05829-f003:**
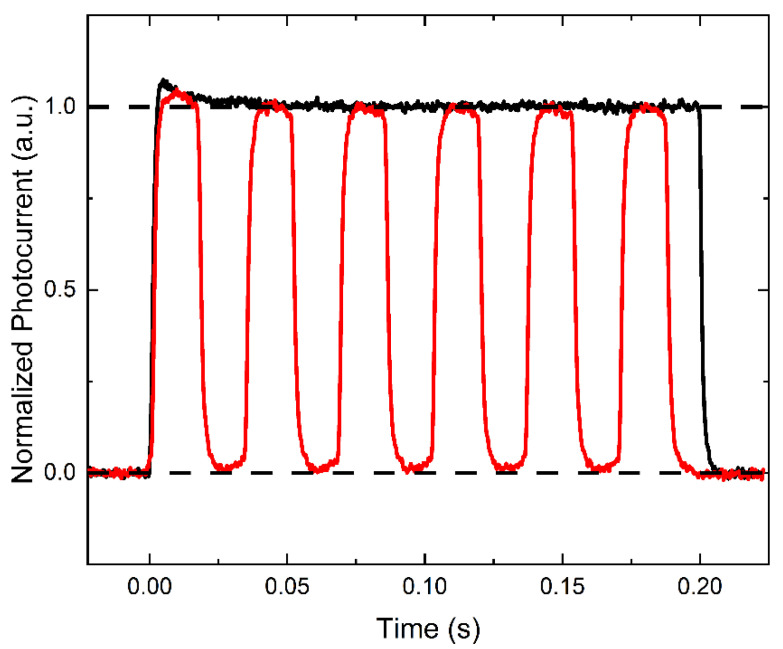
The X-ray response of the PI/a-PbO detector, biased at 20 V/μm, to a continuous (black) and modulated (red) beam of X-rays. Modulated beam has a frame rate of 30 frames per second, matching that used in fluoroscopy. The photocurrent is normalized to the steady-state magnitude of the continuous response.

**Figure 4 sensors-22-05829-f004:**
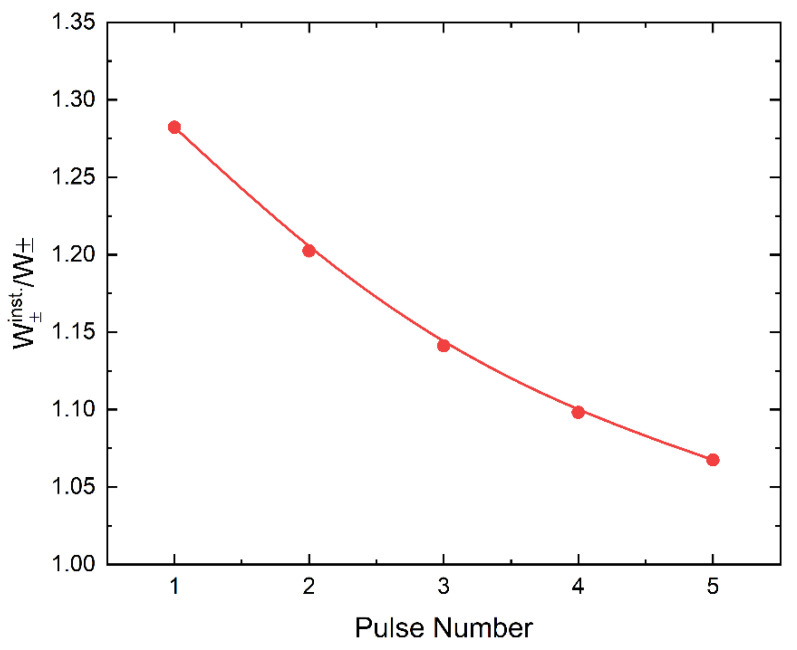
The ratios of W_±_ measured immediately after the application of the bias ($W_{+}^{\mathrm{inst}.}$) to W_±_ measured after waiting 10 min post bias application ($W_{\pm }^{\mathrm{inst}.}/W_{\pm }$). Note that W±, measured 10 min after the bias was applied was chosen as the reference point because it is observed that after this amount of waiting, the ehp creation energy remains relatively constant, undergoing very little change over time.

**Figure 5 sensors-22-05829-f005:**
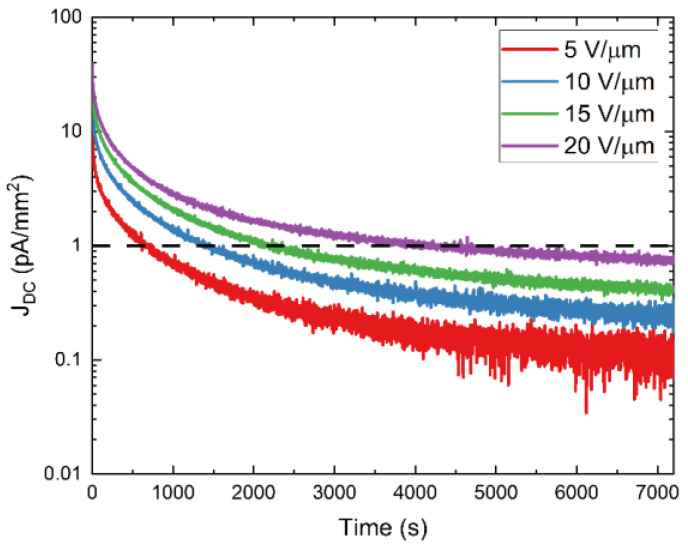
Experimental DC kinetics data plotted in a semi-log scale corresponding to a PI/a-PbO detector biased at selected fields (5–20 V/µm) for two hours. The horizontal dashed line illustrates the operational threshold of 1 pA/mm^2^. Data extracted from [5].

**Figure 6 sensors-22-05829-f006:**
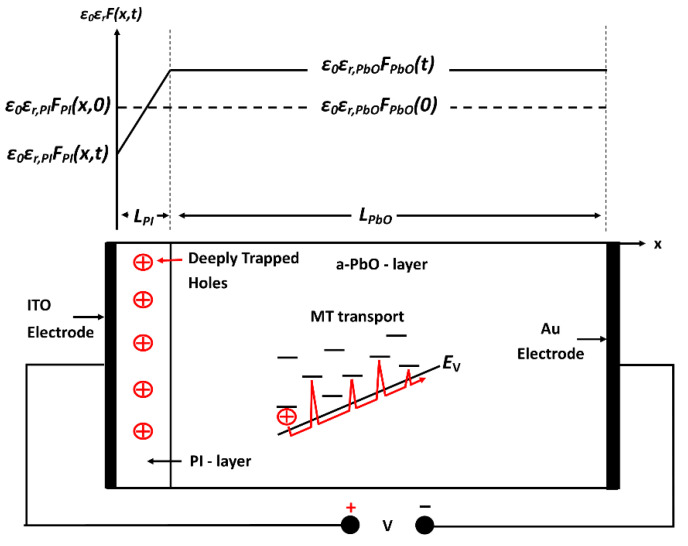
A simplified schematic diagram of the PI/a-PbO detector and its time-dependent spatial electric displacement field profile. The dashed line represents the displacement field at the instant of bias application. The solid line represents the displacement field profile post-bias application when holes have accumulated in PI. In addition, a schematic of MT transport of thermally generated holes through the bulk of a-PbO is illustrated.

**Figure 7 sensors-22-05829-f007:**
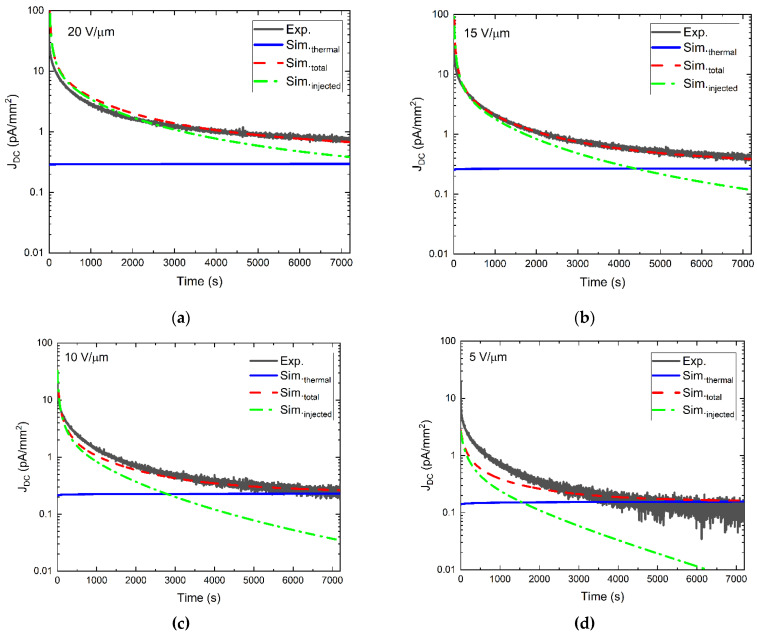
Experimental (solid black) and simulated (injected (dash-dotted green), thermal (solid blue), and total (dashed red)) DC kinetics data plotted in a semi-log scale corresponding to a PI/a-PbO detector biased at fields of (**a**) 20 V/µm, (**b**) 15 V/µm, (**c**) 10 V/µm, and (**d**) 5 V/µm for two hours.

**Figure 8 sensors-22-05829-f008:**
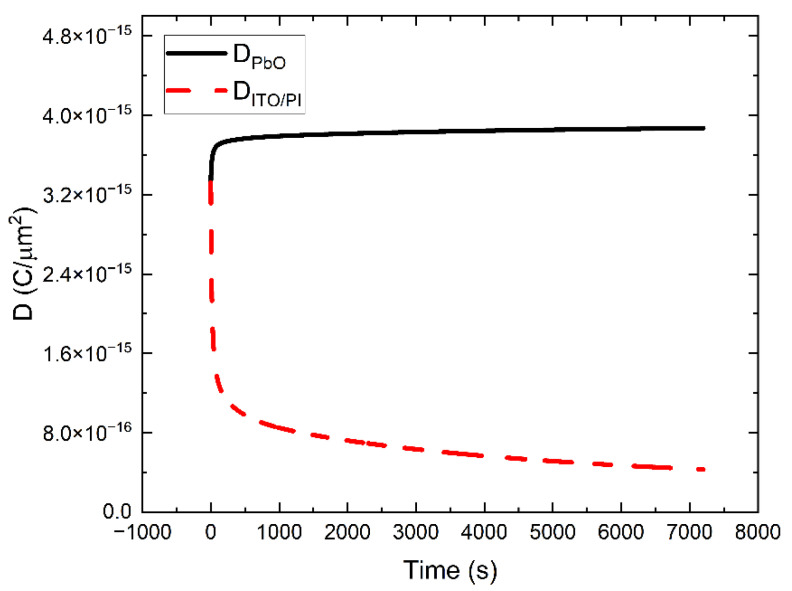
Electric displacement field (D(t)) at the ITO/PI interface (dashed red) and throughout the bulk of a-PbO (solid black) as functions of time for an applied field of 20 V/µm.

**Figure 9 sensors-22-05829-f009:**
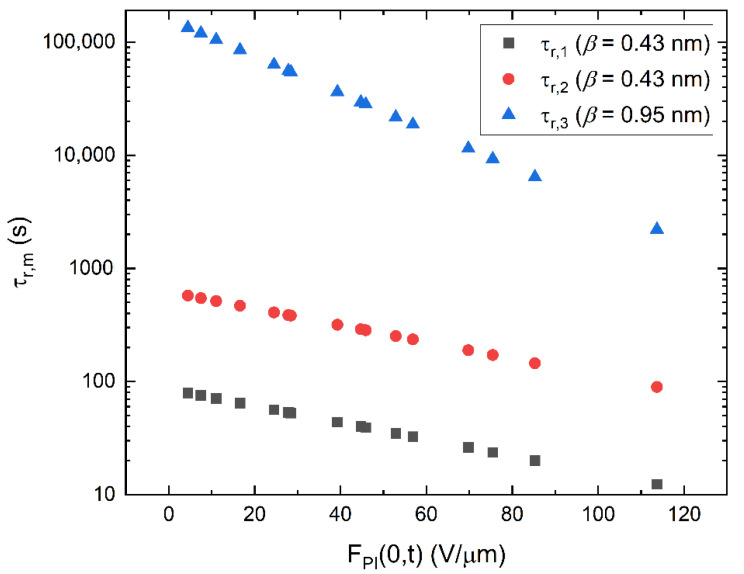
Release time ($\tau_{r,m}$) plotted as a function of the instantaneous field at the ITO/PI interface ($F_{PI}\left(0,t\right)$). For the deepest level of traps at 1.0 eV ($\tau_{r,3}$), a stronger field dependence, compared to the other levels, yielded a more accurate fit between experimental and simulated data.

**Figure 10 sensors-22-05829-f010:**
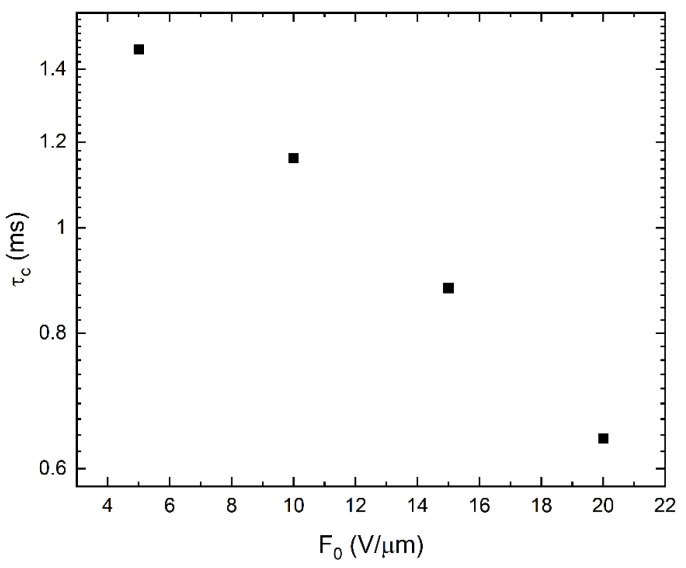
Capture time ($\tau_{c}$) plotted as a function of the applied nominal field (*F*_0_). A unique constant $\tau_{c}$ was given for each applied nominal field to obtain the best fitting between experimental and simulated data.

**Figure 11 sensors-22-05829-f011:**
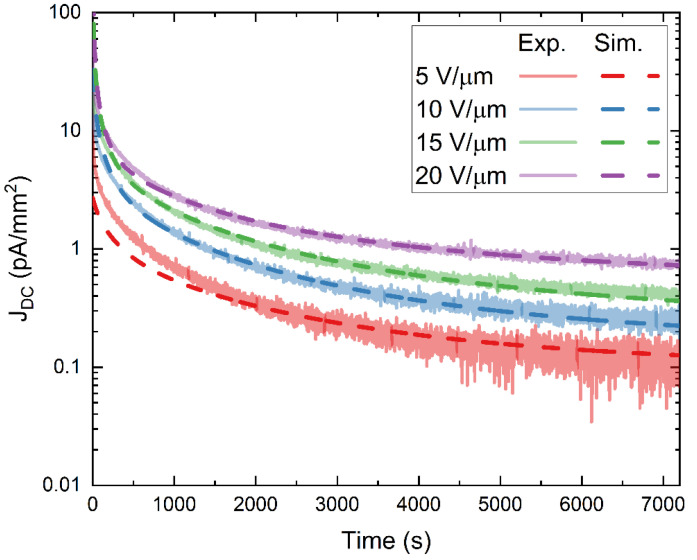
Simulated (dashed lines) and experimental (solid lines) DC kinetics data plotted in a semi-log scale corresponding to a PI/a-PbO detector biased at selected fields (5–20 V/µm) for two hours. Here the model is modified by treating the release times ($\tau_{r,m}$) and capture times ($\tau_{c}$) as electric field dependent parameters in accordance with the hopping enhanced release and capture mechanisms discussed above.

**Figure 12 sensors-22-05829-f012:**
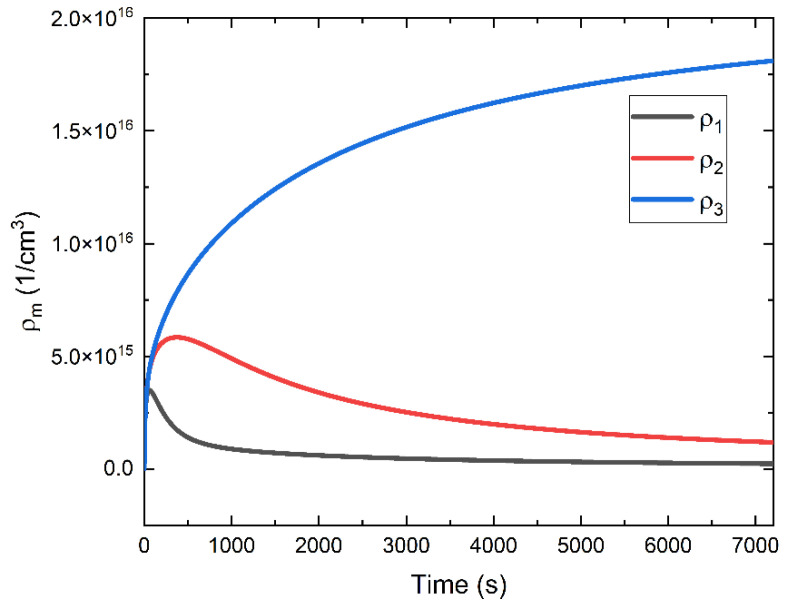
The occupancy of trapping sites, segmented into three discrete levels, plotted as a function of time. Here, these data are obtained from the simulated kinetics corresponding to a nominal field of 20 V/μm. Note that these data are simulated from the unmodified model, where hopping-assisted release and capture is not accounted for.

**Table 1 sensors-22-05829-t001:** Parameters, and their sources, utilized within the mathematical model in this investigation.

Parameter	Value	Source
$\mu_{h}$	1 × 10^−6^ cm^2^V^−1^s^−1^	[15]
$\varepsilon_{r,PI}$	3.3	[32], Dark CELIV
$\varepsilon_{r,PbO}$	26	Dark CELIV
β	0.5 nm	Fitting parameter
$\tau_{c}$	9 × 10^−4^ s	Fitting parameter
$N_{v}$	6 × 10^21^ cm^−3^	[15]
$\omega_{0}$	1 × 10^12^ s^−1^	[6,9]
$g_{h}$	2.32 × 10^11^ cm^−3^s^−1^	Fitting parameters based on [33,34]
$E_{\rho ,m=1}$, $E_{\rho ,m=2}$, $E_{\rho ,m=3}$	0.82, 0.86, 1.0 eV	[15,28]
$N_{\rho ,m=1}$, $N_{\rho ,m=2}$, $N_{\rho ,m=3}$	1.0 × 10^18^, 1.0 × 10^17^, 2.8 × 10^16^ cm^−3^	Fitting parameter
*Ф_h_*	0.81 eV	Fitting parameter
$\tau_{h}$	1.8 × 10^−6^ s	[22]
γ	0.6	[31]
a	0.56 nm	[31]
$E_{a,PbO}$	0.5 eV	[31]

## Data Availability

The data presented in this study are available on request from the corresponding author.

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
