# Peer review of "Dark Current Modeling for a Polyimide—Amorphous Lead Oxide-Based Direct Conversion X-ray Detector"

_sensors, 2022, doi:10.3390/s22155829_

Round 1
Reviewer 1 Report
In the manuscript, the authors analysed and discussed dark current decay over time as a function of bias voltage in Polyimide - Amorphous Lead Oxide- based X-ray Detector. They proposed novel theoretical model to describe the dark current kinetics in an a-PbO detector as a function of bias voltage. It is based on a model previously applied to a-Se photodetectors.
This work is an extension of their earlier publications (Ref. [15] https://doi.org/10.3390/s21217321 and in particular Ref. [4] https://doi.org/10.1109/ TED.2021.3067616).
A look at the aforementioned earlier publications shows that part of the results (Figs. 3 and 5) are merely a reproduction of those previously reported in ref. [4]. However, the second part, in which a new theoretical model is presented, is valuable for publication. It is highly questionable whether the part of the manuscript relating to the modulated and continuous XPM experiment should remain in the manuscript, especially since it is not relevant for the discussion of dark current kinetics and the theoretical model.
The content is clearly stated and the results are adequately discussed. There are several, mostly minor comments and suggestions:
• Fig. 8: I do not see the point of sub-figures a) and b) as they present the same information scaled only for the dielectric constant.
• Conclusion (line 425): "Much better agreement between the experimental and simulated results is evident when compared to Figure 7." should be supported by giving quantitative values for the goodness of fit shown in Fig. 7 and Fig. 11 (e.g. R-squared).
Author Response
Please see the attachment for a point-by-point response.

Reviewer 2 Report
My comments are in attachement

Author Response

(The authors gave the same response as above.)

Round 2
Reviewer 2 Report
The authors' kindly reply to all my curiosities and suggestions, I dont'h have further ones.